# Comparative analysis of the impact of chickenpox and herpes zoster vaccination in Belgium under two different exogenous boosting mechanisms

James Wambua[1]*, John C. Lang[2], Benson Ogunjimi[3,4,5], Niel Hens[1,3], Philippe Beutels[3,6]

1 Data Science Institute, I-BioStat, Hasselt University, Hasselt, Belgium, 2 Merck Canada Inc., Kirkland, Quebec, Canada, 3 Centre for Health Economics Research and Modelling of Infectious Diseases, Vaccine & Infectious Disease Institute, University of Antwerp, Antwerp, Belgium, 4 Antwerp Unit for Data Analysis and Computation in Immunology and Sequencing (AUDACIS), University of Antwerp, Antwerp, Belgium, 5 Department of Paediatrics, Antwerp University Hospital, Antwerp, Belgium, 6 School of Public Health and Community Medicine, The University of New South Wales, Sydney, Australia

* james.wambua@uhasselt.be

## Abstract

### Background

Chickenpox (CP) and herpes zoster (HZ), both caused by the varicella-zoster virus (VZV), present a significant public health burden in unvaccinated populations. Universal CP vaccination has been long debated due to concerns about a potential increase in HZ incidence as a consequence of the exogenous boosting hypothesis.

### Methods

We performed a cost-utility analysis on two deterministic compartmental dynamic transmission models, each of which employed a different underlying mechanism of exogenous boosting: temporal or progressive immunity. We considered four vaccination strategies: the current practice of no widespread vaccination and three strategies involving CP and HZ vaccination, either alone or in combination. The CP vaccines considered were Varivax and ProQuad, while the HZ vaccine considered was the recombinant zoster vaccine (RZV), Shingrix. The vaccine prices per dose were as follows: Varivax - €52.52, ProQuad - €73.69, recombinant zoster vaccine - €170.26. The clinical and economic impact of vaccination on both CP and HZ outcomes were evaluated. The main health outcome of interest was the quality-adjusted life year (QALY) which was used to compare the strategy yielding the highest average net monetary benefits (i.e., the optimal strategy) across a range of willingness-to-pay (WTP) values. Costs and health outcomes were discounted at 3.0% and 1.5% annually,

**Data availability statement:** The social contact datasets utilized in this study are available in the Zenodo-based repository, www.socialcontactdata.org/data. The demographic datasets utilized are available from (statistics Belgium) (STATBEL) https://statbel.fgov.be/nl. All other datasets are presented within the main text or S1 file.

**Funding:** Funding for this project was provided by Merck Sharp & Dohme LLC, a subsidiary of Merck & Co., Inc., Rahway, NJ, USA (MSD) and was received by PB grant number FFP220295 (https://www.merck.com/). NH acknowledges funding from the Methusalem-Centre of Excellence consortium VAX-IDEA. The funders had no role in study design, data collection and analysis, decision to publish, or preparation of the manuscript.

**Competing interests:** JCL is a full-time employee of Merck Canada Inc., Kirkland, Quebec, Canada. JCL may hold stock or stock options in Merck & Co., Inc., Rahway, NJ, USA. BO, NH, and PB are faculty members whose institution (University of Antwerp, Antwerp, Belgium) was paid by MSD. PB reports research grants for unrelated work from Pfizer and AstraZeneca. JW is a voluntary research fellow and a former postdoctoral fellow at the University of Hasselt, Hasselt, Belgium and has no other interests to declare. This does not alter our adherence to PLOS ONE policies on sharing data and materials.

**Abbreviations:** CP, Chickenpox; HZ, Herpes zoster; VZV, Varicella zoster virus; CMI, Cell-mediated immunity; PHN, Postherpetic neuralgia; USA, United States of America; Temp, Temporal immunity boosting; Prog, Progressive immunity boosting; UA, University of Antwerp; MSEIR, Maternal antibody protected–susceptible–exposed–infectious–recovered; VCR, Vaccination coverage rate; INMB, Incremental net monetary benefit; WTP, Willingness-to-pay; QALY, Quality-adjusted life year; PSA, Probabilistic sensitivity analysis; ENLF, Expected net loss acceptability frontier.

respectively. We used 3 time horizons (i.e., 50, 75 & 100 years) and implemented the healthcare payer perspective throughout the analysis.

## Results

CP vaccination led to a substantial reduction in CP incidence in both models. In the temporary immunity boosting (*Temp*) model, strategies that included HZ vaccination showed a decrease in HZ incidence. For the CP vaccination strategy in the *Temp* model, and for all CP and HZ vaccination strategies in the progressive immunity boosting (*Prog*) model, we observed both short- and medium-term increases in HZ, followed by a decrease to levels below the no-vaccination scenario. From the healthcare payer's perspective, using a WTP of €40,000 per QALY gained, the *Temp* model indicated that the three vaccination strategies were cost-effective when considering time horizons of 50, 75, and 100 years. For the *Prog* model, only strategies combining both CP and HZ vaccination were cost-effective given a 100-year time horizon. Vaccination strategies under the *Temp* model became cost-effective at lower values of WTP compared to those under the *Prog* model.

## Conclusion

Both models predicted that universal CP vaccination would result in significant reductions in the burden of CP disease, however, the HZ disease burden impact varied significantly depending on the assumed boosting mechanism. Hence, the choice of modeled exogenous boosting mechanism leads to different optimal vaccination strategies. Ascertaining the relative accuracy of these structural model choices will require continued research on the mechanism of VZV boosting.

## 1. Background

Varicella zoster virus (VZV) is a herpes virus that can result into two clinically different diseases: varicella, also known as chickenpox (CP), and herpes zoster (HZ), also known as shingles [1,2]. CP is a common childhood disease and is characterized by an itching vesicular rash and fever. Primary CP infection is mostly mild, however, it can result in more serious consequences in adults [3]. VZV infection is transmitted from person to person primarily through direct contact with primary skin lesions or through inhalation of aerosolized droplets from acute vesicles of CP or HZ [4]. VZV re-infection is rare in immunocompetent individuals [5]. Following primary VZV infection, VZV remains latent in dorsal root ganglia which can reactivate as HZ, following a decline in VZV-specific cell-mediated immunity (CMI) [1,3,6], more specifically a lack of T cells being capable to recognise specific VZV proteins [6]. HZ is a painful rash that commonly occurs in elderly or immuno-compromized individuals [2]. The incidence and severity of HZ increases with age with more burden occurring in individuals aged ≥50 years [7,8]. HZ can cause several complications, the most common being postherpetic neuralgia (PHN), defined as a painful condition that persists for

more than 3 months after rash onset [9]. Both CP and HZ cause substantial health and economic burden in the population in the absence of universal vaccination [8,10–14]. The annual global burden of CP is estimated to be approximately 140 million cases, with 4.2 million severe cases requiring hospitalization and around 4,200 deaths [12]. A recent systematic review of 69 publications of HZ reported an incidence rate ranging from 5.35 to 10.9 cases per 1,000 person-years, with a recent upward trend [15].

The biological mechanisms underlying pathogenesis of HZ are poorly understood. In 1965, Hope-Simpson came up with the so-called exogenous boosting hypothesis to explore the epidemiological relationship between CP and HZ. This hypothesis posits that secondary exposures to VZV boosts CMI, thereby lowering the reactivation risk of HZ [16]. Although effective vaccines against CP and HZ are available, the implementation of universal vaccination programs is still lacking in many countries [17], primarily because of the unknown drivers of HZ reactivation, particularly the exogenous boosting hypothesis. Previous modeling studies have predicted that CP vaccination will significantly reduce CP incidence but lead to a short- and medium-term increases in HZ incidence due to decreasing VZV exogenous boosting given a decline in CP following vaccination [18,19]. Multiple countries have already implemented universal CP vaccination programs, such as the United States of America (USA), Canada, Germany, Italy, Australia and Israel, each of which have observed a substantial reduction in the CP incidence [20]. The study of HZ incidence in these countries has led to inconclusive findings regarding the role of exogenous boosting. Some countries have seen an increase in HZ incidence following universal CP vaccination, however these trends were also reported prior to the introduction of CP vaccination [13,21]. Nonetheless, the impact of having children in the household (and therefore likely more exogenous boosting events in the household) and immunological studies seemed to confirm the hypothesis [22,23].

Due to the persistent doubts on the quantitative impact of VZV reactivation, mathematical models to assess the impact of universal vaccination programs for CP and HZ have relied on theoretical assumptions to incorporate exogenous boosting mechanisms [22,24]. Two major formulations of the exogenous boosting mechanisms have been considered in mathematical models; temporal (*Temp*) or progressive (*Prog*) immunity boosting [21,22]. Temporal immunity boosting mechanism (*Temp*) [18,25] assumes that boosting provides complete protection for a given duration, after which the individual reverts to full HZ susceptibility [18,25]. The second approach, the progressive immunity boosting mechanism (*Prog*), assumes that CMI accumulates with each VZV re-exposure [19]. These approaches have yielded qualitatively similar but quantitatively different HZ incidence predictions after universal CP vaccination programs were implemented. The quantitative differences are reflected in cost-effectiveness analyses, which are key in making informed decisions on universal vaccination programs [26]. Comparative modeling studies between these two boosting mechanisms are important to understand the uncertainty surrounding policy making based on cost-effectiveness. Qualitative comparisons between model-based and empirically observed post-vaccination incidence patterns of CP and HZ in countries with universal VZV vaccination programs in place, such as the United States of America (USA) are equally important.

With the current study, we aim to compare the effectiveness and cost-effectiveness outcomes of two models of CP and HZ vaccination, one model mimicking a temporary and the other a progressive immunity boosting mechanism using common input data. Throughout this work, we consider the CP vaccines (Varivax and ProQuad) by Merck & Co., Inc., Rahway, NJ, USA (MSD), and recombinant zoster vaccine, Shingrix, by GlaxoSmithKline SA, Belgium.

## 2. Methods

This work results from a collaboration between Merck Sharp & Dohme LLC, a subsidiary of Merck & Co., Inc., Rahway, NJ, USA (MSD), and the University of Antwerp (UA) in Belgium. Both teams independently used a deterministic compartmental modeling framework with different exogenous boosting assumptions to model the relationship between VZV and HZ. Both teams adapted their models to the Belgian context with common input data. Belgium, located in Northwestern Europe, has a population of approximately 11.8 million and an average life expectancy at birth of 82.4 years (2024). The population age structure consists of 16.7% under 15 years, 63.9% between 15 and 64 years, and 19.4% aged 65 years

and above [27]. The MSD team utilized the *Temp* model for the reactivation of VZV to HZ, whereas the UA team utilized the *Prog* model. A detailed description of these two exogenous boosting mechanisms are contained hereafter in Sub-subsections 2.1.1 & 2.1.2. This work, focused on comparing two different transmission models, each with different exogenous boosting mechanisms in accordance with multi-model comparison guidelines [28]. An extensive description of the two model structures and adapted exogenous boosting mechanisms are contained in the S1 File Sections 1 & 2. Four different vaccination strategies were selected for comparison (Subsection 2.4). Each team independently generated model simulations and the UA team performed the comparative analyses of model outputs.

### 2.1. Description of the two exogenous boosting mechanisms

**2.1.1. Progressive immunity boosting mechanism (Prog model).** The *Prog* model assumes that each re-exposure to VZV provides a boost of the individual's CMI leading to a gradual reduction in the risk of reactivation to HZ [19]. The risk of reactivation depends on the individual's age ($a$), the number of previous VZV exogenous exposure events ($i$), and the time elapsed since their last exposure ($\tau$). The *Prog* reactivation rate $\rho_i(a, \tau)$ is modeled as follows.

$$\rho_i(a, \tau) = \rho_0 q^{(i-1)^2} \exp(\theta_a(a - a_0)^+) \exp(\theta_\tau \tau),$$

where $(a - a_0)+ = \max(a - a_0, 0)$, $a_0$ is a threshold when age starts having an influence on the risk of reactivation, $\rho_0$ is a scalar representing the risk of developing HZ for individuals who have recovered from CP, $0 < q < 1$ is a parameter for reduced HZ reactivation risk due to previous VZV exposures, $\theta_a > 0$ is a parameter shaping the increased risk of HZ reactivation with increasing age, and $\theta_\tau > 0$ is the parameter shaping the increased risk of HZ reactivation with time since last exposure, $\tau$. Hence for a given $(a, \tau)$, $\rho_i(a, \tau) > \rho_{i+1}(a, \tau)$ under the progressive immunity mechanism [19]. Based on this functional form for reactivation risk, a discrete deterministic compartmental model based on the **MSEIR** (maternal antibody protected-susceptible-exposed-infectious-recovered) framework for VZV transmission, combined with HZ susceptibility states to describe the reactivation of VZV to HZ, was formulated [19,29]. An extensive description of the flows between compartments is contained in the S1 File.

**2.1.2. Temporal immunity boosting mechanism (Temp model).** The *Temp* model assumes individuals who recover from CP acquire temporal and full immunity against VZV reactivation [25]. However, the acquired protection wanes and individuals become susceptible to HZ after a certain period of time. The risk of VZV reactivation to HZ is formulated as follows.

$$\rho(a) = \omega e^{-\psi a} + a^\eta \pi,$$

where $\omega > 0$, $\psi > 0$, $\eta > 0$, $\pi > 0$ are free parameters of the model [25].

### 2.2. Data description

The datasets utilized in this comparative study are described in the data description Section 4 in S1 File. These include age-stratified CP seroprevalence, HZ incidence, social contact data, mortality rate predictions, population predictions, fertility rate predictions, migration rates, QALY estimates for CP and HZ, CP and HZ general practitioner (CP) consultation rates, CP and HZ hospitalization rates, CP and mortality rates, cost of CP and HZ treatments. In the subsection below, we describe the vaccine prices and administration costs for the vaccines considered in this study.

**2.2.1. Vaccine pricing and cost of administration.** The costs of vaccines were obtained from the Belgian Center for Pharmacotherapeutic report [30] indicating a private sector price per dose and is as follows: recombinant zoster vaccine (GSK) - €170.26; Varivax (MSD) - €52.52; ProQuad (MSD) - €73.69. The costs for the CP vaccine administration per dose

was taken to be €18.16, whilst that for the recombinant zoster vaccine was taken to be €30.0, based on the tariff for an accredited GP consultation in Belgium [31].

### 2.3. Model calibration procedures

Both models were calibrated using age-stratified CP seroprevalence and HZ incidence data. Each model applied a slightly different multi-step approach to optimize the CP-related and HZ-related parameters. A detailed description of these model calibration procedures are contained in Subsections 1.4 & 3.1 in the S1 File for the *Prog* and *Temp* models, respectively.

### 2.4. Vaccination strategies

We considered four vaccination strategies as follows:

1. **$No_{VAC}$ :** the current practice of no widespread CP and HZ vaccination.

2. **$CP_{1y+5y}$ :** routine two-dose CP vaccination at 1 year (vaccination coverage rate (VCR) = 95%) and 5 years (VCR = 90%) with a 4-year catchup single dose CP vaccination at 5 years of age (VCR = 70%).

3. **$CP_{1y+5y} + HZ_R$ :** CP vaccination as in $CP_{1y+5y}$, complemented with HZ vaccination using recombinant zoster vaccine offered routinely at 60 years.

4. **$CP_{1y+5y} + HZ_{R+CU}$ :** CP and routine HZ vaccination as in $CP_{1y+5y} + HZ_R$, complemented with a one-off HZ catch-up vaccination of 61–85 year olds in the year of vaccine implementation.

Both the CP and HZ vaccination programs were set to begin in the year 2023. The ProQuad vaccine was administered at 1 year of age, while the Varivax vaccine was administered at 5 years of age.

### 2.5. Clinical outcomes

Model outputs included: age-stratified CP cases (both wild-type and breakthrough), age-stratified hospitalized CP cases, age-stratified number of CP deaths, age-stratified HZ cases, age-stratified hospitalized HZ cases, age-stratified HZ deaths, and costs and QALYs for each vaccination strategy, age-specific population sizes for the duration considered in the simulation.

### 2.6. Health economic framework

The health economic evaluation was conducted according to Belgian guidelines, which included using the health care payer's perspective, and discounting costs and health outcomes at 3.0% and 1.5% annually, respectively [32]. All costs were adjusted to the 2023 price level using Belgian Consumer Price Indices (see also Subsection 4.17 in the S1 File). We used the incremental net monetary benefit (INMB) to identify the optimal vaccination strategy (i.e., the strategy yielding the highest average net monetary benefit). The INMB values the differences in health effects/outcomes in monetary terms using a willingness-to-pay (WTP) threshold for cost-effectiveness and then subtracts the incremental costs [33,34].

$$INMB = WTP \times \text{Incremental QALYs} - \text{Incremental COSTs}.$$

Since Belgium does not have an explicit official WTP threshold per QALY, we considered €40,000 per QALY as a base case. INMBs were evaluated for time horizons of 50, 75, and 100 years. As a sensitivity analysis, WTP values from €0 to €120,000 were considered. Probabilistic sensitivity analysis (PSA) was used to explore the robustness of our findings in the presence of the underlying uncertainty in the input parameters. Given the differences in epidemiological calibration settings, the *Temp* model accounted for parametric uncertainty in both epidemiological and health economic parameters,

whereas the *Prog* model only did so for the economic parameters. With the PSA, 104 sets of input values were drawn independently from the probability distributions of the input parameters in both models. The values of the input parameters for the cost-utility analyses are contained in Supplementary Table 3 for the *Prog* model, and Supplementary Tables 6 & 7 for the *Temp* model in the S1 File.

## 3. Results

### 3.1. CP incidence

Under the no-vaccination strategy, both models predicted a generally similar and stable overall CP incidence over time (Fig 1). CP incidence in 2023 was 10.45 per 1,000 person-years in the *Prog* model and 10.05 per 1,000 person-years in the *Temp* model. CP incidence stabilized at 9.50 cases per 1,000 person-years in the *Prog* model and ranged between 9.16 and 10.19 per 1,000 person-years in the *Temp* model. A noticeable difference was observed in the cumulative number of CP cases in the *Prog* versus the *Temp* model over a 100-year time horizon. The *Prog* model estimated 12,763,980 CP cases over 100 years (2023–2123), while the *Temp* model estimated 13,611,750 cumulative CP cases (Supplementary Fig 8 in S1 File). CP vaccination resulted in a drastic decline in CP incidence in both models during the first five years following vaccine implementation. CP incidence decreased to approximately 0.002 and 0.35 cases per 1,000 person-years across all vaccination strategies in the *Prog* and *Temp* models, respectively. After 50 years, a slight increase in CP incidence was observed in the *Prog* model, with minor differences between the vaccination strategies. The cumulative number of CP cases across the different vaccination strategies ranged from 290,259–399,547 cases in the *Prog* model and from 645,203–695,473 in the *Temp* model (Supplementary Fig 8 in S1 File). The differences in cumulative CP cases between the *Prog* and *Temp* models were likely due to the differences in model structure and parameterisation regarding vaccination and demographic population flows.

We observed a temporary increase in incidence of primary CP infection in unvaccinated individuals in older age groups in the *Prog* model relative to pre-vaccination era (Supplementary Fig 9 in S1 File). This excess incidence was

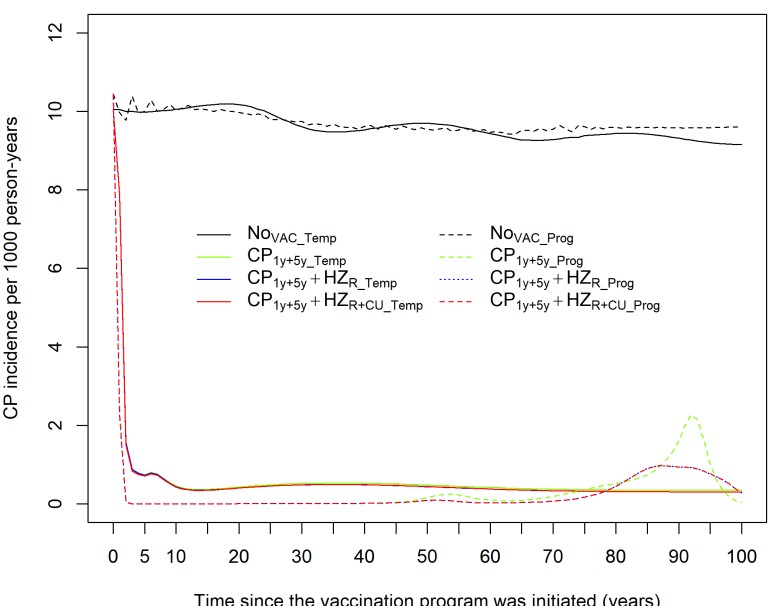

**Fig 1. Model projections for CP incidence across all vaccination strategies over time.** (Dashed) Progressive immunity model (*Prog* model). (Solid) Temporary immunity model (*Temp* model).

only observed in the age groups 15–24 and 25–100 years after more than 80 years after vaccine implementation. There were no apparent increases in the incidence of primary CP infection in the *Temp* model relative to the pre-vaccination era. The observed differences in CP incidence are potentially due to slightly differing assumptions on mechanisms of vaccine protection and waning between the models. See Subsections 1.1 & 2.1 in the S1 File for more information on how CP vaccination is implemented for the *Prog* and *Temp* models, respectively.

### 3.2. HZ incidence

The HZ incidence patterns varied significantly between the two models (Fig 2). For the no vaccination strategy, the incidence of HZ was stable at approximately 4.8 and 5.2 cases per 1,000 person-years for *Prog* and *Temp* models, respectively. The strategy with only CP vaccination ($CP_{1y+5y}$) led to the highest HZ incidence in both models compared to the other strategies. We observed significant differences in HZ incidence between the two models for this strategy in the short- and medium term. In the *Prog* model, the predicted HZ incidence showed an increasing trend, with peak incidence observed approximately 46 years after vaccine implementation, representing a 78.7% increase in incidence compared to the no vaccination strategy. In the *Temp* model, HZ incidence under this strategy increased slightly, with the peak observed 21 years after vaccine implementation, representing a 2.86% increase in incidence compared to the no vaccination strategy. For the vaccination strategies involving both CP and HZ, with or without a catch-up component for HZ (i.e., $CP_{1y+5y} + HZ_R$ and $CP_{1y+5y} + HZ_{R+CU}$), we observed qualitatively different HZ incidence patterns between the two models. In the *Prog* model, we observed increasing HZ incidence, with peak incidence occurring 46 years after vaccine implementation, representing a 39.6% increase in incidence compared to the no vaccination strategy. After this peak, the HZ incidence decreased for 35 years, after which a slight increase in incidence was observed. In the *Temp* model, HZ incidence showed a decreasing trend after vaccine implementation, leveling off at approximately 2.5 cases per 1,000 person-years after eight decades.

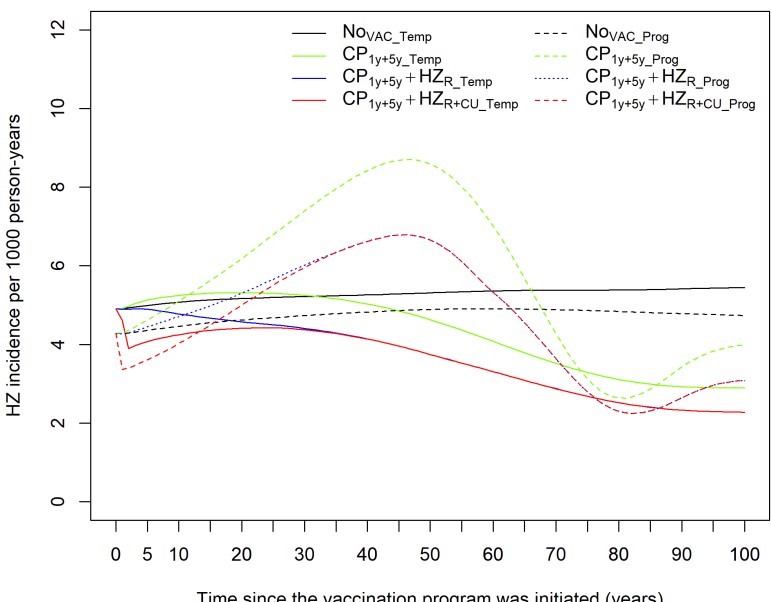

**Fig 2. Model projections for HZ incidence across all vaccination strategies per 1,000 population over time.** (Dashed) Progressive immunity model (*Prog* model). (Solid) Temporary immunity model (*Temp* model).

The cumulative number of HZ cases under the two models was quantitatively different. Under the no vaccination strategy, the *Prog* model predicted 6,267,590 HZ cases over a 100-year time horizon, while the *Temp* model yielded 7,511,947 HZ cases (Supplementary Fig 10 and Supplementary Table 7 in S1 File). The difference of 1,244,357 HZ cases was largely due to differences in the modeled population sizes in the two models and slight variations in the predicted HZ incidence over time (Fig 2). For the other three vaccination strategies, which included either CP vaccination alone or a combination of CP and HZ vaccination, we observed that CP vaccination alone yielded the highest cumulative number of HZ cases in both models. The *Prog* model estimated higher cumulative HZ cases in the three vaccination strategies compared to the *Temp* model. The age-specific HZ incidence in the two models for the different strategies was qualitatively similar but quantitatively different in the age groups 0–14, 15–30, and 31–50 (Supplementary Fig 11 in S1 File). The HZ incidence for the age group 51–100 yielded different qualitative trends and incidences by model. For the age groups under 50 years of age, the *Temp* model produced generally higher HZ incidence compared to the *Prog* model. In contrast, for the age group 51–100 years, the *Prog* model yielded higher HZ incidence across the different vaccination strategies (Supplementary Fig 11 in S1 File).

### 3.3. Health economic evaluation

The estimated QALY loss, total costs, and INMBs are presented in Supplementary Table 8 in S1 File for each strategy and model. Under each of the three time horizons considered (i.e., 50, 75, and 100 years), and for each strategy, the *Prog* model yielded higher QALY losses and total costs than the *Temp* model. The INMBs were higher for the *Temp* model than for the *Prog* model across all strategies. These differences in QALY losses, costs, and INMBs were largely attributable to the exogenous boosting mechanism and the associated higher HZ incidence for the *Prog* model. In the *Temp* model, all three vaccination strategies were found to be cost-effective compared to the no vaccination strategy at a WTP of €40,000, irrespective of time horizon (Fig 3). The CP vaccination strategy alone ($CP_{1y+5y\_Temp}$) was found to be the optimal strategy, having the highest average INMBs across all three time horizons. In the *Prog* model, neither the CP vaccination alone ($CP_{1y+5y\_Prog}$) nor the combined CP and HZ vaccination strategies (i.e., $CP_{1y+5y} + HZ_{R\_Prog}$ and $CP_{1y+5y} + HZ_{R+CU\_Prog}$) were found to be cost-effective when compared to no vaccination for the 50- and 75-year time horizons. However, for the 100-year time horizon, the combined CP and HZ vaccination strategies ($CP_{1y+5y} + HZ_{R\_Prog}$ and $CP_{1y+5y} + HZ_{R+CU\_Prog}$) were found to be cost-effective, with the combined CP and HZ strategy with an initial HZ catch-up ($CP_{1y+5y} + HZ_{R+CU\_Prog}$) being the optimal strategy (Fig 3).

Varying the WTP per QALY from €0 to €120,000, vaccination strategies using the *Temp* model became cost-effective compared to no vaccination at lower WTP values than those under the *Prog* model (Fig 4). In the *Temp* model, CP vaccination alone ($CP_{1y+5y\_Temp}$) emerged as the optimal strategy at the lowest WTP values across the three time horizons. Using the *Prog* model, CP vaccination alone was dominated by the no vaccination strategy across all three time horizons, irrespective of WTP level (Fig 4).

Based on the cost-effectiveness acceptability curves and frontier, we observed that for lower WTP values, the no-vaccination strategy had the highest probability of being cost-effective in both models. However, for higher WTP values, combined CP and HZ vaccination with an initial HZ catch-up had the highest probability of being cost-effective in both models across all three time horizons (Supplementary Fig 12, S1 File). Decision uncertainty was expressed by expected net loss curves (ENLC) and expected net loss acceptability frontier (ENLF) (Supplementary Fig 13 in S1 File). The ENLC showed the consequences of choosing suboptimal vaccination strategies (i.e., the average net loss of choosing a cost-ineffective vaccination strategy) across the range of WTP values considered. Both models yielded different expected net losses for the various vaccination strategies. With the *Temp* model, the no-vaccination strategy ($No_{VAC\_Temp}$) had the highest expected net monetary loss for WTP values exceeding approximately €40,000, €32,000, €26,000 over the 50, 75 and 100-year time horizons, respectively. With the *Prog* model, we found that the CP vaccination strategy alone ($CP_{1y+5y\_Prog}$)

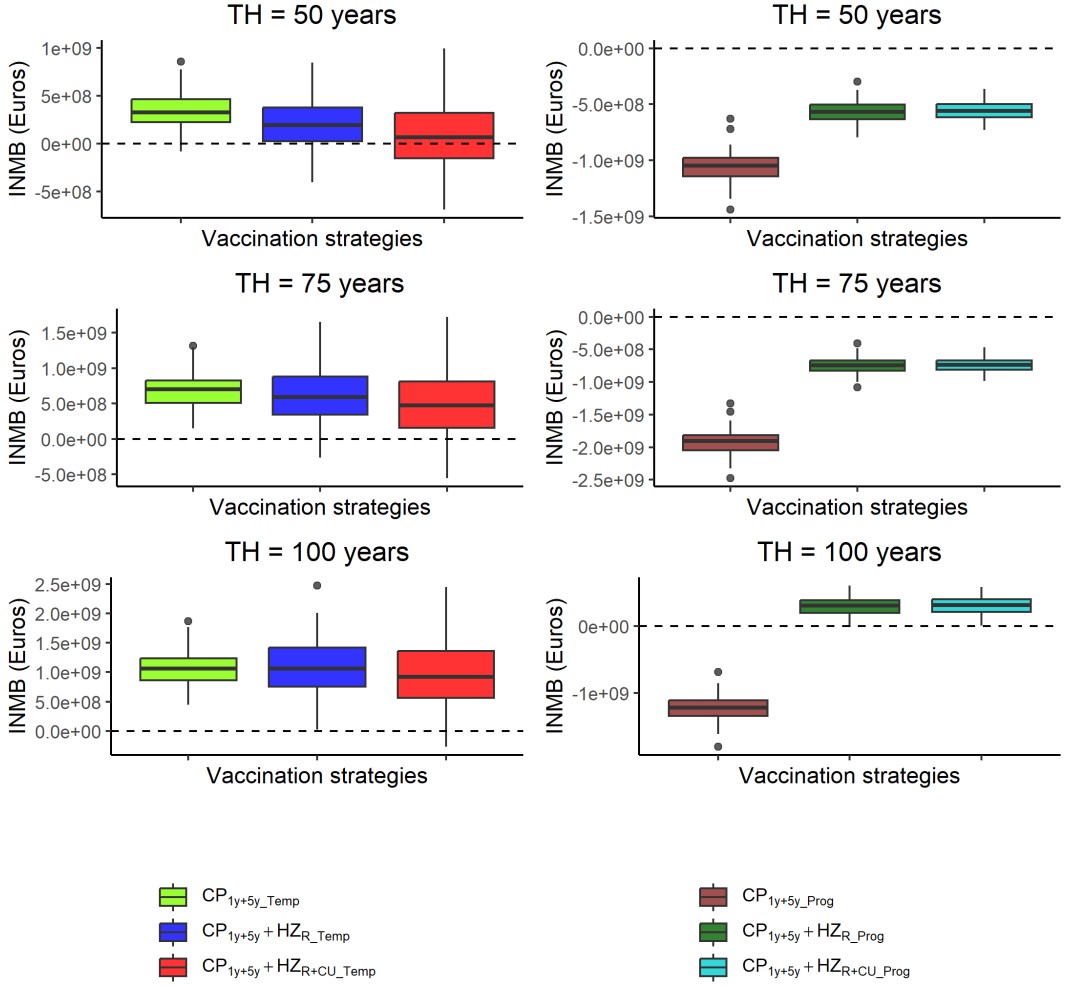

**Fig 3. Box plots for incremental net monetary benefits (INMBs) for the *Temp* model (left) and the *Prog* model (right) with 2.5%, 25%, 50%, 75% and 97.5% quantiles obtained under probabilistic sensitivity analysis, by inter-strategy comparisons, using a WTP threshold of €40,000 per QALY gained.** Three time horizons TH = 50, 75 & 100 years, are considered.

had the highest expected monetary loss for WTP values exceeding approximately €30,000, €22,000, €20,000 over the 50, 75 and 100-year time horizons, respectively.

## 4. Discussion

We compared two independently developed mathematical models, each incorporating a different exogenous boosting mechanism for VZV reactivation to HZ to evaluate the effectiveness and cost-effectiveness of multiple vaccination scenarios in Belgium. We found that CP vaccination significantly reduced CP disease incidence in both models. This result is consistent with the empirical CP incidence observed in countries that have already implemented a universal CP vaccination program [35–41]. The predicted CP incidence in Belgium from each model is also in line with previous predictions from mathematical models formulated to explore the impact of CP vaccination programs [18,26,42–47]. Despite the expected substantial reductions in CP burden following universal vaccination, many countries around the world have delayed implementing universal CP vaccination programs [17]. Several major concerns regarding the delayed

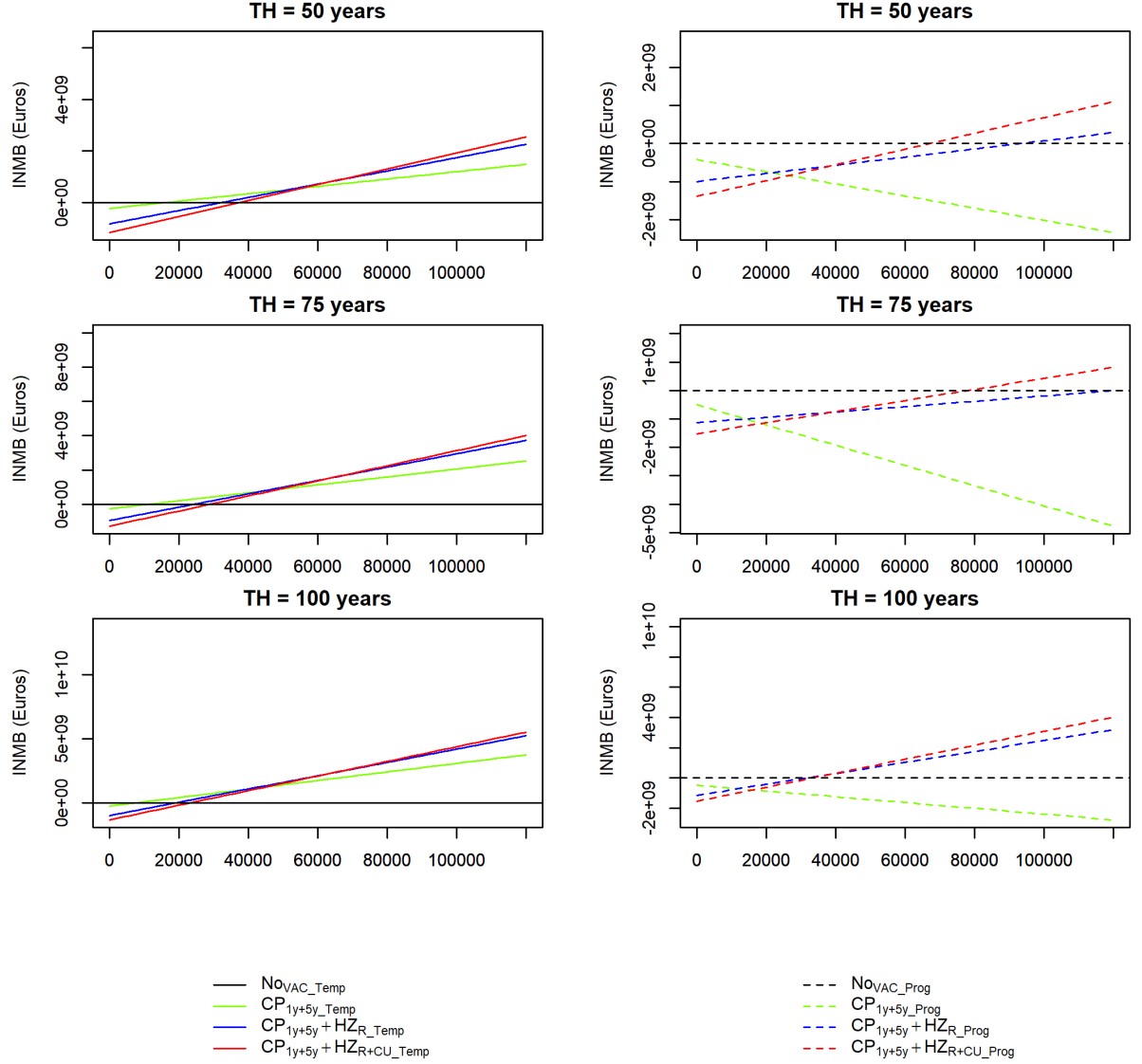

**Fig 4. Expected incremental net benefit (INMBs) for the *Temp* model (left) and the *Prog* model (right) for the different vaccination strategies obtained under probabilistic sensitivity analysis, by inter-strategy comparisons over a range of WTP per QALY values and three time horizons (TH) (50, 75 and 100 years).**

implementation of universal CP vaccination programs include concerns in an age shift of CP to older age groups, which could lead to more severe cases [48]; concerns for increased HZ incidence due to the exogenous boosting hypothesis [16,19,29]; perception of childhood CP as a mild disease [49], and also as a low public health priority, particularly in developing countries [50]. Achieving high CP vaccination coverage (i.e., ≥ 80%) with a CP vaccine catch-up component is vital, as an age shift in CP to older ages could result from low to intermediate vaccination coverage [12]. Here, we assumed that high CP vaccination coverage was achieved (i.e., 95% and 90% for the first and second doses, respectively, and 70% for the one-dose catch-up). Given these assumptions, only the *Prog* model projected a temporary increase in CP incidence in older age groups relative to the pre-vaccination era, with this excess incidence only

observed in the age groups 15–24 and 25–100 years after more than 80 years, i.e., well beyond the time-horizon of observed real world evidence (Supplementary Fig 9 in S1 File). Therefore, both models are consistent with the empirical evidence to date for CP.

After CP vaccine introduction, we observed quantitative differences in HZ incidence over the short- and medium term in the two models. In the *Prog* model, we observed a significant increase in HZ incidence following CP vaccination, whereas in the *Temp* model, only a slight increase in HZ incidence was observed after CP vaccination. The magnitude of the impact of CP vaccination programs on HZ mainly depends on the exogenous boosting mechanism employed [21]. Studies that have utilized the progressive immunity framework [26,29,47,51,52] to model the exogenous boosting mechanism have predicted a higher impact of universal CP vaccination programs on HZ incidence compared to those that have employed the temporary immunity boosting framework [18,42,44–46,53], consistent with the results observed here. In the long-term, we observed comparable HZ incidences in the two models, which were noticeably lower than those of the no vaccination strategies. This can be attributed to the decrease in the number of adults harboring dormant wild-type VZV due to CP vaccination during childhood, along with a reduced risk of reactivation of the VZV vaccine-strain to HZ [54,55]. The strategies that included both CP and HZ vaccination, either with or without initial HZ catch-up mitigated the increased HZ incidence observed in the *Temp* model. In contrast, although the *Prog* model with HZ vaccination showed a significant reduction in HZ incidence in the short- and medium term compared to CP vaccination alone, it remained higher than the no vaccination strategy.

Cost-effectiveness outcomes of the two models were driven by the projected incidence of HZ, which is associated with higher morbidity and mortality than CP [8,56]. The *Temp* model, which revealed a minimal influence of CP vaccination on HZ incidence when compared to the *Prog* model, yielded more favorable cost-effectiveness outcomes at lower willingness-to-pay (WTP) values. These findings are consistent with previous studies that have modeled the cost-effectiveness of CP vaccination while incorporating the two exogenous boosting assumptions, as highlighted in a recent systematic review [57]. Here, the CP vaccination only strategy produced markedly different cost-effectiveness outcomes between the two models. In the *Temp* model, this strategy emerged as the optimal approach at lower WTP values. Conversely, in the *Prog* model, CP vaccination alone was consistently dominated by the no vaccination strategy. These discrepancies were primarily attributable to larger adverse effects of CP vaccination on HZ incidence in the *Prog* model. The inclusion of HZ vaccination alongside CP vaccination in the combination strategies completely mitigated the increased HZ incidence in the *Temp* model, whereas it only partially mitigated the increase in HZ incidence in the *Prog* model. Consequently, the strategies that included both CP and HZ vaccination became optimal in both models at higher WTP values, with variations in the WTP values depending on the model and time horizon considered. The substantial differences in both effectiveness and cost-effectiveness between the two models underscore the extent of the underlying uncertainties regarding the mechanisms of VZV reactivation to HZ. Therefore, to conclusively elucidate the impact of CP vaccination on HZ incidence, further research into these underlying mechanisms is needed.

To attain a more profound understanding of the extent and magnitude of the exogenous boosting hypothesis postulated several decades ago, real-world evidence is imperative [21,22]. A multidisciplinary systematic review conducted by Ogunjimi et al. [22] aimed to investigate the risk reduction of HZ through exposure to CP patients. This review concluded that exogenous boosting does exist, albeit not in all individuals or circumstances, and its magnitude remains inadequately quantified across various research domains, particularly due to limitations related to insufficient follow-up duration and study heterogeneity [22]. An updated review by Talbird et al. [21] sought to summarize new evidence on exogenous boosting. Overall, this updated review corroborated the findings of Ogunjimi et al. regarding the existence of exogenous boosting. Hence, given the epidemiological plausibility of exogenous boosting, supported by both epidemiological and immunological studies as highlighted in the two reviews, exposure to VZV can, at times, mitigate the risk of HZ. Thus, it is crucial to investigate the magnitude and duration of individual-level protection, as well as the extent of population-level effects [22,58].

A self-controlled case series study conducted in the UK to explore the risk of HZ after exposure to VZV found that in adults, the risk of HZ within 2 years of exposure to a CP infected child in the household was 33% lower than in the baseline period. Furthermore, the study indicated that in the 10–20 years after exposure, the risk of HZ was 27% lower, suggesting that re-exposure to VZV provides some protection from the risk to HZ, but not complete immunity [59]. An empirical study by Ogunjimi et al. [23], concluded that boosting only occurred in 17–25% of grandparents who were exposed to their CP infected grand child and that this boost lasted for less than one year. An individual-based study integrating both within-host and between-host VZV-CMI data to explore HZ incidence estimated the duration of exogenous boosting to be 2 years [60]. Findings from these studies suggested reduced boosting and protective effects at individual-level following VZV re-exposure, compared to previous predictions that suggested it might last 20 years [14]. Therefore, it can be argued that a re-evaluation of prior mathematical modeling studies could be needed, as the majority had assumed an average duration of protection of 20 years [14]. The population-level impact of exogenous boosting remains inadequately understood. In the aforementioned updated review by Talbird et al. [21], 14 of the 22 observational studies that provided HZ incidence data both pre- and post-CP vaccine implementation generally reported increasing rates of HZ incidence over time in both the pre- and post-universal CP vaccination phases. Additionally, the rates of HZ incidence across age groups did not demonstrate consistent differences, hence making it difficult to ascribe the HZ incidence increases to CP vaccination alone versus other factors. A recent systematic review and meta-analysis performed to explore the impact of CP vaccination on HZ incidence with a focus on population-level effects revealed an increase in age-adjusted HZ incidence before implementation of CP vaccination which did not change thereafter [61]. A recent study in USA by Leung et al. comparing HZ incidence in birth cohorts preceding and following CP vaccination program launch, concluded that empirical data do not support earlier model predictions suggesting that the program would increase HZ incidence among persons who previously experienced CP [58]. Nonetheless, it is noteworthy that this study also found a greater rise in HZ incidence in adults living with children relative to those living without children in the post-vaccination era [58]. A population-based cohort study conducted in Minnesota, USA to explore the temporal changes in HZ incidence using data from 1945–1960 and 1980–2007 showed that the HZ incidence increased > 4-fold over that period with no change in the rate of increase pre- and post introduction of the CP vaccination program [62]. Another study in USA found no difference in HZ incidence in states with low versus high CP vaccination coverage rates [63]. A recent study in Germany showed the rate of increase in HZ-related hospitalizations has remained steady following the launch of the CP vaccination program [64]. Other real-world studies in countries with no CP vaccination program such as the UK have shown an increased rate of HZ incidence over time [13]. Hence, it could be alluded that the re-exposure of VZV to HZ susceptible individuals does not have noticeable population-level impact on HZ incidence because of the limited individual-level boosting effects and short-lived duration of protection [58]. However, since previous studies indicated that factors such as demographic changes (i.e., aging of the population and changing fertility rates) [26,47], comorbidity of chronic illnesses [65], psychological stress and immunosuppressive drugs [66], could have a subtle impact in the observed increase in HZ incidence, more research on the association of these factors and HZ incidence is crucial both in countries with and without universal CP vaccination programs. This is key in gaining more insights on the role of CP vaccination on HZ incidence.

The lack of clear understanding on the role of exogenous boosting in CP and HZ epidemiology initiates inherent difficulties in exploring the relationship between CP and HZ and thus creates considerable uncertainty from the different modeling hypotheses currently employed in the literature. Our comparative study used both the temporal and progressive immunity boosting mechanisms. The validity of either of these boosting mechanisms is still debated as the underlying biological/immunological mechanisms are largely unknown to date. Thus, more studies elucidating the underlying biological mechanisms of the exogenous boosting hypothesis as well as understanding the HZ pathogenesis are needed in order to gain more insights on the relationship between VZV and HZ [19,21,22]. Additionally, there is need for mathematical models to explore the VZV and HZ transmission dynamics in countries with high quality data on age-specific HZ incidence both before and after CP vaccination programs, and also high quality data on social contact patterns. Such models could

facilitate the comparison of model predictions and the available observed incidence data. This could shed more light on whether there is real-world evidence of exogenous boosting at population level [21].

This work has several limitations. First, we utilized social contact data collected before the COVID-19 pandemic in Belgium and assumed constant social contact patterns in our model predictions. The pandemic seems to have initiated changes in contact patterns which may have important implications on the epidemiology of many pathogens, including CP and HZ [67]. Second, although we took into consideration demographic changes in both models through predicted estimates for fertility and mortality for the Belgian population, these predictions are characterized by different assumptions. Third, the two models had slightly different assumptions about CP vaccine protection, leading to small differences in the long-term incidence and cumulative CP cases. Fourth, substantial underlying uncertainty persists regarding the risk of HZ due to reactivation of the vaccine-strain, owing to limited data available for evaluating the relative incidence of HZ in individuals vaccinated against CP. This uncertainty is further compounded by the fact that existing data are predominantly collected from young individuals and over a limited time span [55]. Furthermore, there is limited data to explore the average duration of vaccine-induced immunity against both CP and HZ in vaccinated individuals. Clearly, more studies are needed in the above areas. A fundamental limitation of standard health economic analyses is that the implied utilitarian ethical framework under which policy makers seek to maximise the total sum of QALYs in a population under a budget constraint is not the only possible framework. For instance, contractualism could serve as an alternative to utilitarianism, and the accurate quantification of VZV boosting effects then becomes much less important. This could lead to prioritising the protection of children through CP vaccination despite increases of HZ incidence in elderly [68].

## 5. Conclusion

The predicted impact of universal CP vaccination on HZ incidence is dependent on how exogenous boosting is implemented in dynamic transmission models. Different implementations can lead to significant differences in the cost-effectiveness of CP vaccination. Policy makers should be aware that model predictions for CP and HZ may depend heavily on the exogenous boosting mechanism adopted, which should be taken into account when evaluating CP and HZ vaccination policies. Finally, additional research is needed to determine which mechanism of exogenous boosting is most likely and to explore other plausible mechanisms.

## Supporting information

**S1 File. Details on the dynamic transmission models employed, equations governing the models, model calibration procedures, epidemiological and health economic parameter values, dataset description, and Supplementary Figures and Tables.**
(PDF)

## Acknowledgments

We would like to thank Kevin Bakker for reading through the manuscript and for his input.

## Author contributions

**Conceptualization:** James Wambua, John C. Lang, Benson Ogunjimi, Niel Hens, Philippe Beutels.

**Data curation:** James Wambua, John C. Lang.

**Formal analysis:** James Wambua, John C. Lang.

**Funding acquisition:** Niel Hens, Philippe Beutels.

**Investigation:** James Wambua, John C. Lang, Niel Hens, Philippe Beutels.

 

**Methodology:** James Wambua, John C. Lang, Benson Ogunjimi, Niel Hens, Philippe Beutels.

**Project administration:** John C. Lang, Philippe Beutels.

**Software:** James Wambua, John C. Lang.

**Supervision:** John C. Lang, Niel Hens, Philippe Beutels.

**Visualization:** James Wambua.

**Writing – original draft:** James Wambua.

**Writing – review & editing:** James Wambua, John C. Lang, Benson Ogunjimi, Niel Hens, Philippe Beutels.

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
