## [Decision Letter · Decision Letter 0]

30 Oct 2025

Dear Dr. Wambua,

Thank you for submitting your manuscript to PLOS ONE. After careful consideration, we feel that it has merit but does not fully meet PLOS ONE’s publication criteria as it currently stands. Therefore, we invite you to submit a revised version of the manuscript that addresses the points raised during the review process.

We look forward to receiving your revised manuscript.

Kind regards,

Georges M.G.M. Verjans, MSc, PhD

Academic Editor

PLOS ONE

Journal Requirements:

JCL is a full-time employee of Merck Canada Inc., Kirkland, Quebec, Canada. JCL may hold stock or stock options in Merck & Co., Inc., Rahway, NJ, USA. BO, NH, and PB are faculty members whose institution (University of Antwerp, Antwerp, Belgium) was paid by MSD. PB reports research grants for unrelated work from Pfizer and AstraZeneca. JW is a voluntary research fellow and a former postdoctoral fellow at the University of Hasselt, Hasselt, Belgium and has no other interests to declare.

Additional Editor Comments :

MS reviewed by two senior scientists, one VZV expert and the other an acknowledged computational and mathematical modeler, who both considered the study of interest but had significant concerns that need to be addressed. The revised MS will be reviewed by the same reviewers.

Reviewers' comments:

Reviewer's Responses to Questions

**Comments to the Author**

1. Is the manuscript technically sound, and do the data support the conclusions?

Reviewer #1: Yes

Reviewer #2: No

2. Has the statistical analysis been performed appropriately and rigorously?

Reviewer #1: Yes

Reviewer #2: N/A

3. Have the authors made all data underlying the findings in their manuscript fully available?

Reviewer #1: Yes

Reviewer #2: Yes

4. Is the manuscript presented in an intelligible fashion and written in standard English?

Reviewer #1: Yes

Reviewer #2: Yes

Reviewer #1: This manuscript includes analyses of varicella and herpes zoster vaccination strategies in the country of Belgium, in order to determine the various effects on herpes zoster burden. The authors observed that the impact varied significantly depending on the assumed boosting mechanism of immunity to VZV over the lifespan of an individual. The manuscript is filled with data, including considerable Supplemental Data. However, the interpretations of the data are not always clearly delineated within the manuscript. Further the costs of herpes zoster vaccination are buried within the Supplemental Data. The costs of vaccination should be clearly stated in the main text of the manuscript. See comments below.

1.Title. To this Reviewer, it seems peculiar to call the varicella vaccine by the name of chickenpox vaccine? The Merck Company even named their vaccine as Varivax (not chickenvax).

2.Herpes zoster vaccine in title and everywhere in the manuscript. The are 2 herpes zoster vaccines: the live vaccine (Zostavax) and the recombinant vaccine Shingrix (often called RZV). From the title, it is unclear which vaccine is being examined. Shingrix only cited later in line 176 of Methods. Suggest adding the word recombinant zoster vaccine into the title to avoid confusion.

3.Abstract. Likewise, in Abstract, add the word recombinant to zoster vaccine.

4.Method. The title states that the study uses data from Belgium. Restate that point at the front of Methods. Also give us basic data about Belgium: the average life span, total population, percent of population under 18 years, and percent of population over 65 years.

5.Methods, line 171. Add the word recombinant zoster vaccine for clarity.

6.Methods. New section needed. Section 4.19 in Supplemental data called Vaccine Pricing. THESE DATA ARE CRITICAL. Move this entire paragraph on vaccine costs into the main text of Methods as a new section.

7.Discussion. Line 323. Clearly re-state in the first few sentences of the Discussion that this analysis pertains only to the country of Belgium. Or if that statement is not true, clarify whether this analysis can be applied to other countries, and which other countries. The life span in Belgium is around 82 years. By comparison, the life span in at least 10 countries in Africa is under 60 years.

8.Major discrepancy with a published article. In the Supplemental data, the authors state that the cost of Shingrix is 170 (vaccine) + 30 (administration) = 200 Euros. In a paper in the journal VACCINE (Vol 30:675, 2012; PMID: 22120193), Bilcke et al state the cost of a zoster vaccine must be less than 45 Euros to be cost-effective in Belgium. Please cite this paper in the Discussion and explain the different costs and values for QALY in the 2 reports.

9.Abstract. The cost of vaccination is a critical component of this analysis. Please add the cost of vaccination into the Abstract of this manuscript.

10.Conclusion. Again, please clearly state in the Conclusion whether the analysis in this report refers only to Belgium (see comment 7). Avoid use of words such as structural assumptions; say what you mean precisely. Overall, the Conclusion is a weak statement.

Reviewer #2: The article by Wambua and colleagues is a comparative analysis of chickenpox and herpes zoster vaccination in Belgium under two different exogenous boosting mechanisms. This work is important as universal chickenpox and herpes zoster present a major public health problem. The impact of vaccination varies depending on the assumed boosting mechanism. The paper presents a cost-utility analysis on two deterministic compartmental models each employing a different boosting mechanism (temporal or progressive immunity). The models, which have been developed by different teams, investigated the epidemiological impact and cost-effectiveness. The article reported that universal chickenpox vaccination can significantly reduce chickenpox disease burden. However, the impact on herpes zoster disease varies depending on the assumed boosting mechanism. Although the paper includes some technical details in the main text, it is well written. The model is extensive and well described. I have some concerns related to the economic analysis.

Major comments

1. The analysis only use fixed values for costs and QALYs. The paper should investigate the impact of the various costs and QALYs using a sensitivity analysis in which the chosen costs and QALYs are varied with e.g. 50%. In my view this is important as for instance the some of the costs are taken from a paper published in 2012 (reference 16 in the supplement). It is unlikely that the same costs have to be paid in the present day. I therefore believe it is important to include a sensitivity analysis in which these costs are increased (this could for instance be done using tornado plots).

2. Similarly, the QALY scores that are used are from older people whereas the models include the entire life span (these scores are from references 19 and 21 in the supplement). Reference 16 uses different scores. As QALY scores are key for cost-effectiveness, it is important to show the impact of varying these scores on the main outcome.

3. Please add the time horizon and discounting to the abstract as these are important for the outcomes.

Minor comments

1. Please add a label to the y-axis of Figures 3 and 4. Please also use the same scale for all y-axes in Figures 3 and 4.

2. Supplement section 1.1 includes the following sentence: “The risk of HZ reactivation is modeled using the progressive immunity boosting mechanism as described in Subsection ?? in the main text.” Please replace ?? with the appropriate sub-section.

**Do you want your identity to be public for this peer review?** For information about this choice, including consent withdrawal, please see our Privacy Policy

Reviewer #1: No

Reviewer #2: No

---

## [Author Response · Author response to Decision Letter 1]

27 Jan 2026

Our responses to the comments received during our initial submission are all included in the 'Response to Reviewers' file. Thank you.

---

## [Editor Report · Decision Letter 1]

29 Jan 2026

Comparative analysis of the impact of chickenpox and herpes zoster vaccination in Belgium under two different exogenous boosting mechanisms

PONE-D-25-45382R1

Dear Dr. Wambua,

We’re pleased to inform you that your manuscript has been judged scientifically suitable for publication and will be formally accepted for publication once it meets all outstanding technical requirements.

Kind regards,

Georges M.G.M. Verjans, MSc, PhD

Academic Editor

PLOS One

Additional Editor Comments (optional):

The MS has been improved by addressing the reviewers' comments appropriately.
---

## [Editor Report · Acceptance letter]

PONE-D-25-45382R1

PLOS One

Dear Dr. Wambua,

I'm pleased to inform you that your manuscript has been deemed suitable for publication in PLOS One. Congratulations! Your manuscript is now being handed over to our production team.

Kind regards,

on behalf of

Prof. Dr. Georges M.G.M. Verjans

Academic Editor

PLOS One